# Unpaired Point Cloud Completion via Unbalanced Optimal Transport

**Taekyung Lee** [1]   **Jaemoo Choi** [2]   **Jaewoong Choi** [* 3]   **Myungjoo Kang** [* 1 4]

## Abstract

Unpaired point cloud completion is crucial for real-world applications, where ground-truth data for complete point clouds are often unavailable. By learning a completion map from unpaired incomplete and complete point cloud data, this task avoids the reliance on paired datasets. In this paper, we propose the *Unbalanced Optimal Transport Map for Unpaired Point Cloud Completion (UOT-UPC)* model, which formulates the unpaired completion task as the (Unbalanced) Optimal Transport (OT) problem. Our method employs a Neural OT model learning the UOT map using neural networks. Our model is the first attempt to leverage UOT for unpaired point cloud completion, achieving competitive or superior performance on both single-category and multi-category benchmarks. In particular, our approach is especially robust under the class imbalance problem, which is frequently encountered in real-world unpaired point cloud completion scenarios. The code is available at https://github.com/LEETK99/UOT-UPC.

## 1. Introduction

The three-dimensional (3D) point cloud is a fundamental representation in 3D geometry processing (Guo et al., 2020). However, acquiring complete point cloud data of real-world objects remains a significant challenge due to the limitations of the scanning process (Yuan et al., 2018). Consequently, diverse approaches have been proposed for point cloud completion, which aims to reconstruct a complete (full) point cloud from incomplete (partial) data (Yu et al., 2021; Wang et al., 2022; Tchapmi et al., 2019; Chen et al.,

*Equal contribution [1]IPAI (Interdisciplinary Program in Artificial Intelligence, Seoul National University) [2]Georgia Institute of Technology [3]Sungkyunkwan University [4]Department of Mathematical Sciences and RIMS, Seoul National University. Correspondence to: Jaewoong Choi <jaewoongchoi@skku.edu>, Myungjoo Kang <mkang@snu.ac.kr>.

*Proceedings of the 42nd International Conference on Machine Learning*, Vancouver, Canada. PMLR 267, 2025. Copyright 2025 by the author(s).

2020; Hong et al., 2023). Existing approaches can be categorized into paired (supervised) and unpaired (unsupervised) methods. The paired methods rely on training data that explicitly aligns incomplete point clouds with their corresponding complete versions (Yu et al., 2021; Wang et al., 2022; Tchapmi et al., 2019; Xia et al., 2021; Zhou et al., 2021). However, acquiring such paired datasets can be both expensive and challenging. To address these limitations, the **unpaired point cloud completion** has emerged. These approaches aim to train a completion model from the independently sampled incomplete and complete point clouds without explicit one-to-one correspondence. This is achieved by leveraging shared semantic information, such as object class (Ma et al., 2023; Chen et al., 2020; Wen et al., 2021), or through domain adaptation using paired synthetic data (Liu et al., 2024). However, existing unpaired approaches primarily rely on heuristic techniques without a rigorous theoretical formulation of the unpaired point cloud completion problem.

In this paper, we formulate this unpaired point cloud completion through the **Optimal Transport** Map (OT Map) problem. The OT Map is defined as the cost-minimizing transport map that bridges two probability distributions. Recently, several works proposed methods for learning the OT Map using neural networks (Neural OT) (Rout et al., 2022; Fan et al., 2022; Choi et al., 2024a). These models have been applied to various machine learning tasks, such as generative modeling (Choi et al., 2023) and image-to-image translation (Fan et al., 2022; Gazdieva et al., 2024). A key component of Neural OT is the cost function, which determines how each input $x$ is transported to $T(x)$. However, existing approaches mainly focus on variants of the quadratic cost function $l_2$ (Rout et al., 2022; Fan et al., 2022; Choi et al., 2024a; 2023). To address this limitation, we analyze various candidate cost functions for unpaired completion task and show that this theoretical analysis closely aligns with experimental results.

Based on this OT Map formulation, we propose a novel unpaired point cloud completion model based on the Unbalanced Optimal Transport (UOT) framework. We refer to our model as the *Unbalanced Optimal Transport Map for Unpaired Point Cloud Completion (UOT-UPC)*. Our experiments demonstrate that UOT-UPC achieves state-of-the-art performance on unpaired point cloud completion

benchmarks across both single-category and multi-category settings. Furthermore, UOT-UPC exhibits particularly robust performance under class imbalance, where incomplete and complete distributions consist of multiple categories in different proportions. The UOT framework provides our model with inherent robustness against class imbalance, further enhancing its effectiveness in real-world scenarios. Our contributions are summarized as follows:

- UOT-UPC is the first unpaired point cloud completion model based on the Unbalanced Optimal Transport framework, formulating the task as finding the optimal transport map.
- We provide a comprehensive analysis of suitable cost functions and prove a strong alignment between theoretical insights and empirical results.
- UOT-UPC attains state-of-the-art performance on both single-category and multi-category benchmarks.
- UOT-UPC shows robust performance under severe class imbalance problem between incomplete and complete point clouds.

**Notations and Assumptions** Let $\mathcal{X}, \mathcal{Y}$ be two compact complete metric spaces, $\mu$ and $\nu$ be probability distributions on $\mathcal{X}$ and $\mathcal{Y}$, respectively. $\mu$ and $\nu$ are assumed to be absolutely continuous with respect to the Lebesgue measure. Throughout this paper, we denote the source distribution as $\mu$ and the target distribution as $\nu$. Since the focus of this paper is on point cloud completion, $\mu$ **and** $\nu$ **represent the distributions of the incomplete and complete point clouds**, respectively. For a measurable map $T$, $T_{\#}\mu$ represents the pushforward distribution of $\mu$. $\Pi(\mu, \nu)$ denote the set of joint probability distributions on $\mathcal{X} \times \mathcal{Y}$ whose marginals are $\mu$ and $\nu$, respectively. Additionally, $f^*$ indicates the convex conjugate of a function $f$, i.e., $f^*(y) = \sup_{x \in \mathbb{R}}\{\langle x, y \rangle - f(x)\}$ for $f : \mathbb{R} \to [-\infty, \infty]$.

## 2. Preliminaries

**Optimal Transport** The *Optimal Transport (OT)* problem investigates the task of transporting the source distribution $\mu \in \mathcal{P}(\mathcal{X})$ to the target distribution $\nu \in \mathcal{P}(\mathcal{Y})$. This problem was initially formulated by Monge (1781) using a deterministic transport map $T : \mathcal{X} \to \mathcal{Y}$ such that $T_{\#}\mu = \nu$:

$$C(\mu, \nu) := \inf_{T_{\#}\mu = \nu}\left[\int_{\mathcal{X}} c(x, T(x))d\mu(x)\right]. \quad (1)$$

Intuitively, Monge's OT problem explores the optimal transport map $T^*$ that connects two distributions while minimizing a given cost function $c(x, T(x))$. Although Monge's OT problem offers an intuitive framework, it has theoretical limitations: this formulation is non-convex and the optimal transport map $T^*$ may not exist depending on the conditions on $\mu$ and $\nu$ (Villani et al., 2009). To address these issues,

Kantorovich introduced a relaxed formulation of the OT problem (Kantorovich, 1948). Formally, this Kantorovich formulation is expressed in terms of a coupling $\pi$ rather than a transport map $T$, as follows:

$$C_{ot}(\mu, \nu) := \inf_{\pi \in \Pi(\mu, \nu)}\left[\int_{\mathcal{X} \times \mathcal{Y}} c(x, y)d\pi(x, y)\right]. \quad (2)$$

where $c$ is a cost function and $\pi \in \Pi(\mu, \nu)$ is a coupling of $\mu$ and $\nu$. In contrast to the Monge problem, the minimizer $\pi^*$ of Eq 2 always exists under some mild assumptions on $(\mathcal{X}, \mu)$, $(\mathcal{Y}, \nu)$ and the cost function $c$ (Villani et al., 2009). Note that under our assumptions that $\mu$ and $\nu$ are absolutely continuous with respect to the Lebesgue measure, the optimal transport map $T^*$ exists and the optimal coupling is given by $\pi^* = (Id \times T^*)_{\#}\mu$ (Villani et al., 2009).

Rout et al. (2022); Fan et al. (2023) proposed a method for learning the optimal transport map $T^*$ using the semi-dual formulation of OT. This neural network-based approach for learning the OT Map is referred to as *Neural Optimal Transport (Neural OT)*. In specific, these models parametrize the potential function $v$ and the transport map $T$ as follows:

$$\mathcal{L}_{v_\phi, T_\theta} = \sup_{v_\phi}\left[\int_{\mathcal{X}} \inf_{T_\theta}\left[c\left(x, T_\theta(x)\right) - v_\phi\left(T_\theta(x)\right)\right]d\mu(x)\right.$$
$$\left. + \int_{\mathcal{X}} v_\phi(y)d\nu(y)\right]. \quad (3)$$

**Unbalanced Optimal Transport** The classical OT problem assumes an exact transport between two distributions $\mu$ and $\nu$, i.e., $\pi_0 = \mu, \pi_1 = \nu$. However, this exact matching constraint results in sensitivity to outliers (Balaji et al., 2020; Séjourné et al., 2022) and vulnerability to class imbalance in the OT problem (Eyring et al., 2024). To mitigate this issue, a new variation of the OT problem is introduced, called *Unbalanced Optimal Transport (UOT)* (Chizat et al., 2018; Liero et al., 2018b). Formally, the UOT problem is expressed as follows:

$$C_{uot}(\mu, \nu) = \inf_{\pi \in \mathcal{M}_+(\mathcal{X} \times \mathcal{Y})}\left[\int_{\mathcal{X} \times \mathcal{Y}} c(x, y)d\pi(x, y)\right.$$
$$\left. + D_{\Psi_1}(\pi_0|\mu) + D_{\Psi_2}(\pi_1|\nu)\right], \quad (4)$$

where $\mathcal{M}_+(\mathcal{X} \times \mathcal{Y})$ denotes the set of positive Radon measures on $\mathcal{X} \times \mathcal{Y}$. $D_{\Psi_1}$ and $D_{\Psi_2}$ represents two $f$-divergences generated by convex functions $\Psi_i$, and are defined as $D_{\Psi_i}(\pi_j|\eta) = \int \Psi_i\left(\frac{d\pi_j(x)}{d\eta(x)}\right)d\eta(x)$. These $f$-divergences penalize the discrepancies between the marginal distributions $\pi_0, \pi_1$ and $\mu, \nu$, respectively. Hence, **in the UOT problem, the two marginal distributions are softly matched to** $\mu, \nu$, i.e., $\pi_0 \approx \mu$ and $\pi_1 \approx \nu$. Intuitively,

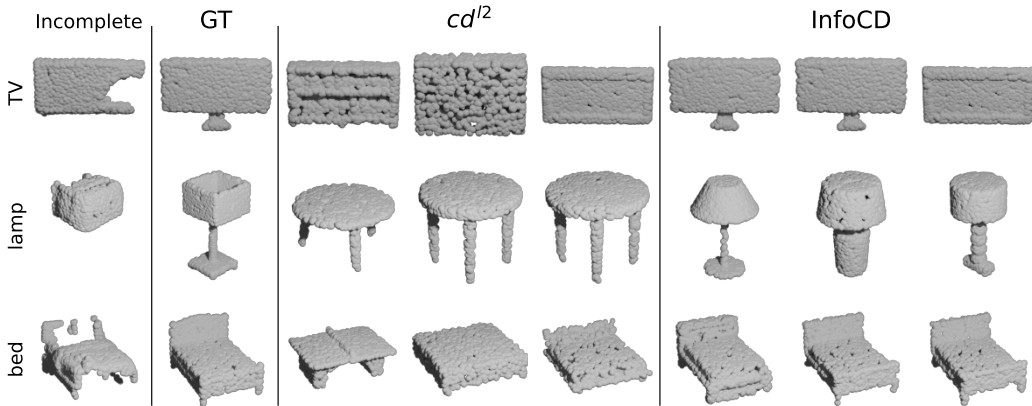

Figure 1: **Visualization of the incomplete point cloud** $x$**, the ground-truth completion** $y^{gt}(x)$**, and three complete point clouds** $y_i^c(x)$ that minimize the cost $c(x, y_i^c(x))$ for two cost functions: $cd^{l2}$ and InfoCD, in the **multi-category setting**.

the UOT problem can be seen as the OT problem between $\pi_0 \approx \mu$ and $\pi_1 \approx \nu$, rather than between the exact distributions $\mu$ and $\nu$ (Choi et al., 2023). This flexibility offers robustness to outliers (Balaji et al., 2020) and adaptability to class imbalance problem between $\mu$ and $\nu$ (Eyring et al., 2024) to the UOT problem (See Sec 4.2 for details). We refer to the optimal transport map $T^\star$ from $\pi_0$ to $\pi_1$ as the **unbalanced optimal transport map (UOT Map)**. Note that, under our assumption that the source and target distributions are absolutely continuous, the existence of this UOT Map is guaranteed (Liero et al., 2018a, Thm. 3.3).

Choi et al. (2023) introduced a Neural OT model for the UOT problem into generative modeling, called UOTM (See Sec 4.2 for details). In this paper, we adapt the UOT Map to the task of unpaired point cloud completion. Unlike generative modeling, in unpaired point cloud completion, each incomplete source sample $x$ should be transported to its corresponding complete target sample $y$. Therefore, it is important to set an appropriate cost function $c(x, y)$ in Eq 4, because this cost governs how each $x$ is transported to $y$. In Sec 4.1, we compare diverse cost functions to ensure that the UOT Map works as a valid completion model.

based model that maps the latent features of the incomplete point cloud to those of the complete point cloud. Wu et al. (2020) proposed a conditional GAN model that generates multiple plausible complete point clouds, conditioned on the incomplete point cloud. ShapeInv (Zhang et al., 2021) employed an optimization-based GAN-inversion approach (Xia et al., 2022). ShapeInv finds the optimal input noise to reconstruct the complete point cloud from the given incomplete point cloud. This is conducted by minimizing the distance between the input incomplete point cloud, which is for completion, and the partial point cloud, which is obtained by degrading the generator's output. Cycle4 (Wen et al., 2021) proposed two cyclic transformations between the latent spaces of incomplete and complete point clouds, utilizing missing region coding. USSPA (Ma et al., 2023) proposed a symmetric shape-preserving method based on GAN.

Unlike prior heuristic-driven methods, our work is the first to provide a theoretical formulation of unpaired point cloud completion as the OT Map problem. Based on this formulation, we propose a theoretically grounded UOT approach for unpaired point cloud completion.

## 3. Related Works

Point cloud completion has been investigated through various approaches. The paired (supervised) methods leverage explicit correspondences between incomplete and completion point clouds to train their models, such as ASFM-Net (Xia et al., 2021) and PVD (Zhou et al., 2021). In contrast, the unpaired (unsupervised) approaches suggest methods that do not rely on paired data. In this regard, unpaired point cloud completion is a more general and challenging problem.

Unpaired (Chen et al., 2020) is one of the first approaches for unpaired point completion. This model introduced a GAN-

## 4. Method

In this paper, our key idea is to **train our model to learn the UOT Map from the incomplete point cloud distribution** $\mu$ **to the complete point cloud distribution** $\nu$**.** In Sec 4.1, we formulate unpaired point cloud completion as an (U)OT Map problem and present an extensive comparison to identify the most suitable cost function. In Sec 4.2, we introduce our max-min learning objective. In Sec 4.3, we provide implementation details, such as training algorithm.

Table 1: **Cost function evaluation by comparing the cost-minimizer** $g_1^c(x)$ **and the ground-truth completion** $y^{gt}(x)$ **for each incomplete point cloud** $x$**.** We evaluate the suitability of each cost function for UOT-UPC by measuring the L1 Chamfer distance ($cd^{l1} \times 10^2 (\downarrow)$) between $g_1^c(x)$ and $y^{gt}(x)$.

(a) Multi-category

| Cost Function | AVG | chair | table | trash bin | TV | cabinet | bookshelf | sofa | lamp | bed | tub |
|---|---|---|---|---|---|---|---|---|---|---|---|
| USSPA | 8.64 | **7.40** | 8.88 | **9.13** | 8.70 | 11.48 | 7.61 | 6.52 | 10.01 | 8.72 | 8.30 |
| $l_2$ | 23.97 | 12.52 | 31.21 | 29.17 | 26.65 | 22.29 | 22.96 | 20.51 | 24.64 | 27.03 | 21.80 |
| $cd^{l2}$ | 9.78 | 8.07 | 7.69 | 14.00 | 5.91 | 18.86 | 7.88 | 7.34 | 6.23 | 8.76 | 7.07 |
| $cd^{l2}_{fwd}$ | 8.87 | 9.48 | 8.62 | 9.38 | 7.80 | **10.55** | 7.73 | **5.63** | 14.59 | 10.32 | 7.28 |
| InfoCD | **8.46** | 7.43 | **6.41** | 11.69 | **5.69** | 17.35 | **6.52** | 6.25 | **2.70** | **6.91** | **4.92** |

(b) Single-category

| Cost Function | AVG | chair | table | trash bin | TV | cabinet | bookshelf | sofa | lamp | bed | tub |
|---|---|---|---|---|---|---|---|---|---|---|---|
| USSPA | 7.18 | 7.44 | 7.15 | 6.98 | 6.08 | 10.02 | 7.00 | 6.12 | 8.35 | 7.90 | 4.79 |
| $l_2$ | 14.88 | 11.21 | 12.52 | 22.37 | 8.29 | 20.46 | 17.87 | 8.69 | 11.57 | 19.55 | 7.07 |
| $cd^{l2}$ | 6.65 | 7.17 | 7.35 | 8.35 | 5.46 | 10.59 | 5.77 | 6.39 | 3.70 | 6.46 | 5.28 |
| $cd^{l2}_{fwd}$ | 6.12 | 7.29 | 7.41 | 7.23 | **5.18** | 9.03 | 6.45 | **4.64** | 2.82 | 6.75 | **4.44** |
| InfoCD | **5.58** | **6.84** | **5.90** | **6.91** | 5.29 | **7.86** | **4.37** | 5.75 | **2.72** | **5.78** | 4.51 |

Table 2: **Class imbalance in the USSPA benchmark dataset (Ma et al., 2023).** The Incomplete and Complete rows indicate the proportion of each class in the respective datasets. The Ratio represents the proportion ratio (incomplete/complete). A Ratio $\neq 1$ indicates the presence of class imbalance.

| class | chair | table | trash bin | TV | cabinet | bookshelf | sofa | lamp | bed | tub |
|---|---|---|---|---|---|---|---|---|---|---|
| Incomplete | 43% | 21.3% | 8.0% | 6.4% | 6.0% | 6.1% | 3.9% | 1.1% | 2.9% | 1.2% |
| Complete | 22.2% | 22.2% | 1.9% | 6.1% | 8.7% | 2.5% | 17.6% | 12.9% | 1.3% | 4.7% |
| Ratio | 1.94 | 0.96 | 4.21 | 1.05 | 0.69 | 2.44 | 0.22 | 0.09 | 2.23 | 0.26 |

## 4.1. Motivation

**Task Formulation as OT Map** We begin by defining our target task: *Unpaired point cloud completion*. Consider two sets of point cloud data: the incomplete set $X = \{x_i \mid x_i \in \mathcal{X}, i = 1, \cdots, N\}$ and the complete set $Y = \{y_j \mid y_j \in \mathcal{Y}, j = 1, \cdots, M\}$. Note that $X$ and $Y$ are unpaired, i.e., $X$ and $Y$ are independently sampled from $\mu$ and $\nu$. In real-world scenarios, obtaining incomplete-complete pairs of point clouds is prohibitively expensive, making this unpaired approach essential. For instance, one can train a model on incomplete real-world objects with complete synthetic data, then perform completion on real-world inputs (Ma et al., 2023). Formally, our objective is to train a completion model $T$ from the unpaired datasets:

$$T : \mathcal{X} \to \mathcal{Y} \quad x \mapsto T(x). \tag{5}$$

where $x$ and $T(x)$ denote the input incomplete point cloud and its corresponding completion. This completion model $T$ must satisfy the following two conditions:

(i) $T$ should generate a complete point cloud sample.

(ii) $T$ should transport each incomplete point cloud to its appropriate complete counterpart $y$.

In this regard, the optimal transport map (Eq. 1) is suitable for the point completion model. By definition, the optimal transport map $T^\star$ is (1) a generator of the complete point

cloud samples, i.e., $T(x) \sim \nu$ for $x \sim \mu$ that (2) optimally minimizes the given cost function $c(x, T(x))$. Therefore, condition (i) is inherently satisfied. The key remaining question is:

Q. Can we induce the OT Map to satisfy condition (ii) by selecting an appropriate cost function $c(\cdot, \cdot)$?

If we can identify such cost function $c(\cdot, \cdot)$ that induces an explicit bias in $T^\star$ to satisfy condition (ii), then $T^\star$ can serve as the point cloud completion model.

**Cost Function Comparison** By definition, the OT Map $T^\star$ is the cost minimizer among all target distribution generators (Eq. 1). Hence, in order to satisfy condition (ii), **the chosen cost function** $c(\cdot, \cdot)$ **should assign a lower cost to** $c(x, T(x))$ **when** $T(x)$ **is the correct completion of** $x$ and a higher cost to $c(x, y)$ when $y$ is not the correct corresponding completion. In this regard, we test various cost function candidates, including $l_2$, $L2$-Chamfer distance ($cd^{l2}$) (Fan et al., 2017), one-directional $L2$-Chamfer distance ($cd^{l2}_{fwd}$), and InfoCD (Lin et al., 2024). For an incomplete (partial) point cloud $x_i = \{x_{in} \in \mathbb{R}^3\}$ and complete point cloud $y_j = \{y_{jm} \in \mathbb{R}^3\}$, each cost function is defined as follows:

- $l_2(x_i, y_j) = \sum_n \|x_{in} - y_{jn}\|_2^2$.

- $cd^{l2}(x_i, y_j)$
  $= \sum_n \min_m \|x_{in} - y_{jm}\|_2^2 + \sum_m \min_n \|x_{in} - y_{jm}\|_2^2$.

- $cd_{fwd}^{l2}(x_i, y_j) = \sum_n \min_m \|x_{in} - y_{jm}\|_2^2$.

- $\text{InfoCD}(x_i, y_j) = \ell_{\text{InfoCD}}(x_i, y_j) + \ell_{\text{InfoCD}}(y_j, x_i)$. where $\ell_{\text{InfoCD}}(x_i, y_j)$
$= -\frac{1}{|y_j|} \sum_m \log \left\{ \frac{\exp\left\{ -\frac{1}{\tau'} \min_n d(x_{im}, y_{jn}) \right\}}{\sum_m \exp\left\{ -\frac{1}{\tau} \min_n d(x_{im}, y_{jn}) \right\}} \right\}$

For each partial point cloud $x$ and a given cost function $c$, we select $k$-nearest complete samples $y_i^c(x)$ for $1 \le i \le k$ based on $c(x, \cdot)$ on the target (completion) dataset. Then, we compare them with the ground-truth completion $y^{gt}(x)$. Our goal is to evaluate each cost function by testing whether the $k$-nearest neighbor $y_i^c(x)$ is indeed similar to the ground-truth completion $y^{gt}(x)$. If so, this suitable cost function can be exploited to train our OT-based completion model via the optimal transport map. The experiment is conducted on paired completion data from ShapeNet (Chang et al., 2015).

Fig. 1 visualize the incomplete point cloud $x$, the ground-truth completion $y^{gt}(x)$, and the 3-nearest neighbor $y_3^c(x)$ for the $cd^{l2}$ and InfoCD cost functions (See Appendix B for additional results for the single-category setting). Fig. 1 show that **selecting the cost-minimizing pair based on** InfoCD **retrieves an appropriate** $y_3^c(x)$**, which closely resembles** $y^{gt}(x)$, in the multi-category setting. Table 1 presents similar results. Table 1 reports the L1 chamfer distance between $y^{gt}(x)$ and the nearest neighbor $y_1^c(x)$ for each cost function. The results indicate that the $l2$ cost performs the worst. This result shows that $l_2$ cost is unsuitable for the point cloud completion task. In contrast, the InfoCD achieves competitive results, performing comparably or better than USSPA on the majority of datasets. **Therefore, in Sec 4.2, we propose an OT Map approach using the InfoCD cost function for the point cloud completion task, based on our investigation of the most suitable cost function**. Furthermore, we conduct an ablation study on the cost function in Sec 5.3 to demonstrate how this cost function comparison closely aligns with the completion performance of UOT-UPC.

**UOT Map for Class Imbalance Problem** In this paragraph, we clarify the motivation for adopting the UOT Map, instead of the classical OT Map. Our goal is unpaired point cloud completion, where the training data $X$ and $Y$ are not provided as paired samples. This inherently introduces the **class imbalance problem**. For instance, consider point cloud data consisting of the 'Chair' and 'Table' categories, where incomplete and complete point clouds originate from different sources, such as real-world scans and synthetic datasets. In this case, the ratio of these two classes may differ between the incomplete point cloud distribution $\mu$ and the complete point cloud distribution $\nu$. For example, while the incomplete point cloud data might consist of 50% 'Chair' and 50% 'Table,' the complete point cloud data could be composed of 70% 'Chair' and 30% 'Table.'

Unfortunately, the standard OT problem (Eq. 1) is susceptible to this class imbalance problem (Eyring et al., 2024). The standard OT Map transports each source sample $x \sim \mu$ to a target sample $y \sim \nu$ without rescaling. Consequently, in this class imbalance case, 20% of the 'Table' incomplete point cloud data would be transported to 20% of the 'Chair' complete point cloud. This behavior is undesirable for a completion model. In practice, **this class imbalance problem is prevalent in the unpaired point cloud completion benchmark (Table 2)**. The proportion of some categories, e.g., 'lamp' and 'trash bin' classes, significantly differs by more than threefold between the incomplete and complete datasets. To address this, we propose using the UOT Map as our point cloud completion model. **The robustness of UOT to class imbalance will be explained in Sec 4.2 and empirically validated through experiments in Sec 5.2.**

### 4.2. Proposed Method

In this section, we introduce our unpaired point cloud completion model, which is based on the UOT Map, called **UOT-UPC**. Our approach is to learn the UOT Map $T^\star$ from the incomplete point cloud distribution $\mu$ to the complete point cloud distribution $\nu$ using a neural network $T_\theta$. To achieve this, we adopt the UOTM framework (Choi et al., 2023), which is based on the semi-dual formulation (Eq. 6) of the UOT problem (Vacher & Vialard, 2023)).

$$C_{uot}(\mu, \nu) = \sup_{v \in \mathcal{C}} \left[ \int_{\mathcal{X}} -\Psi_1^*(-v^c(x))) \, d\mu(x) \right. \\ \left. + \int_{\mathcal{Y}} -\Psi_2^*(-v(y)) d\nu(y) \right], \quad (6)$$

where the $c$-transform of $v$ is defined as $v^c(x) = \inf_{y \in \mathcal{Y}} (c(x, y) - v(y))$. We refer to the optimal maximizer $v^\star$ of Eq. 6 as the optimal potential function for the UOT problem. Following prior approaches for learning the optimal maps (Korotin et al., 2021; Fan et al., 2022; Rout et al., 2022; Choi et al., 2023), we introduce $T_\theta$ to approximate the UOT Map $T^\star$ as follows:

$$T_\theta(x) \in \underset{y \in \mathcal{Y}}{\arg\inf} \left[ c(x, y) - v(y) \right] \\ \Leftrightarrow \quad v^c(x) = c(x, T_\theta(x)) - v(T_\theta(x)), \quad (7)$$

Note that the UOT Map $T^*$ satisfies the above conditions (Eq. 7) with the optimal potential $v^\star$ (Choi et al., 2023). By parametrizing the optimal potential $v^\star$ with a neural network $v_\phi$ and substituting $v^c$ using the right-hand side of Eq. 7, we arrive at the following learning objective $\mathcal{L}_{v_\phi, T_\theta}$:

$$\mathcal{L}_{v_\phi, T_\theta} \\ = \inf_{v_\phi} \int_{\mathcal{X}} \Psi_1^* \left( -\inf_{T_\theta} \left[ c(x, T_\theta(x)) - v_\phi(T_\theta(x)) \right] \right) d\mu(x) \\ + \int_{\mathcal{Y}} \Psi_2^* \left( -v_\phi(y) \right) d\nu(y). \quad (8)$$

---

**Algorithm 1** Training algorithm of UOT-UPC

---

**Require:** The mixture of the incomplete and complete point cloud distribution $\mu$. The complete point cloud distribution $\nu$. $\Psi_i^*(x) = \text{Softplus}(x)$. Generator network $T_\theta$ and the discriminator network $v_\phi$. $dl$ is density loss. Total iteration number $K$.

1: **for** $k = 0, 1, 2, \ldots, K$ **do**
2:     Sample a batch $X \sim \mu, Y \sim \nu$.
3:     $\mathcal{L}_T = \frac{1}{|X|} \sum_{x \in X} c(x, T_\theta(x)) - v_\phi(T_\theta(x)) + dl(T_\theta(x))$
4:     Update $\theta$ by minimizing the loss $\mathcal{L}_T$.
5:     $\mathcal{L}_v = \frac{1}{|X|} \sum_{x \in X} \Psi_1^*(-c(x, T_\theta(x)) + v_\phi(T_\theta(x))) + \frac{1}{|Y|} \sum_{y \in Y} \Psi_2^*(-v_\phi(y))$
6:     Update $\phi$ by minimizing the loss $\mathcal{L}_v$.
7: **end for**

---

Note that the learning objective $\mathcal{L}_{v_\phi, T_\theta}$ reduces to the standard OT Map when $\Psi_i^*$ is the identity function (Eq. 3), which is equivalent to setting $\Psi_i$ as the convex indicator function at $\{1\}$. Hence, the UOT Map is a generalization of the OT Map. Furthermore, the UOT Map can be interpreted as the OT Map between the rescaled distributions $\pi_0(x) = \Psi_1^{*\prime}(-v^{\star c}(x))\mu(x)$ and $\pi_1(y) = \Psi_2^{*\prime}(-v^\star(y))\nu(y)$, where $v^\star$ denotes the optimal potential (Choi et al., 2023). These rescaling factors $\Psi_i^{*\prime}(\cdot)$ offer the **flexibility of the UOT map to handle the class imbalance problem, which is a key challenge in unpaired point cloud completion (Eyring et al., 2024).**

### 4.3. Implementation Details

As described in Algorithm 1, $\mathcal{L}_{v_\phi, T_\theta}$ can be computed by the Monte Carlo approximation with mini-batch samples from the incomplete point cloud $x$ and the complete point cloud $y$. Intuitively, our learning objective is similar to the adversarial training in GANs (Goodfellow et al., 2020). Our potential $v_\phi$ and completion model $T_\theta$ play similar roles as the discriminator and generator in GANs, respectively. This is because the minimization with respect to $T_\theta$ in Eq 8 is equivalent to the maximization of $\mathcal{L}_{v_\phi, T_\theta}$[1].

We parametrize the generator and discriminator using the similar backbone network as USSPA (Ma et al., 2023) (See Appendix A for the implementation details). InfoCD$(\cdot, \cdot)$ (Lin et al., 2024) is adopted as the cost function $c(\cdot, \cdot)$ in the learning objective $\mathcal{L}_{v_\phi, T_\theta}$. Moreover, in practice, we set the source distribution $\tilde{\mu}$ as a mixture of the incomplete point cloud distribution $\mu$ and complete point cloud distribution $\nu$, with a mixing probability of 50%, i.e., $\tilde{\mu} = 0.5\mu + 0.5\nu$. Then, we train the UOT Map between $\tilde{\mu}$ and $\nu$. This mixture trick helps our generator to produce high-fidelity complete

---

[1]Since we assume $\Psi_i$ to be convex and non-negative, its convex conjugate $\Psi_i^*$ is an increasing function.

point clouds. **We conducted ablation studies on the mixture trick and the cost function in Sec 5.3.**

## 5. Experiments

In this section, we evaluate our model from various perspectives. For implementation details of experiments, please refer to Appendix A.

- In Sec 5.1, we evaluate our model on the unpaired point cloud completion benchmarks, considering both single-category and multi-category settings.

- In Sec 5.2, we demonstrate our model's robustness to the class imbalance problem.

- In Sec 5.3, we conduct ablations studies to investigate the effect of different cost functions and the source mixture trick.

### 5.1. Unpaired Point Completion Benchmark

We assess our UOT-UPC model on the unpaired point cloud completion benchmarks under two settings: (1) *Real Data Completion (USSPA dataset (Ma et al., 2023))* and (2) *Synthetic Data Completion (PCN dataset (Yuan et al., 2018))*. For quantitative evaluation, we measure the L1 Chamfer distance (Fan et al., 2017) $(cd^{l1})$ and F-scores (Tatarchenko et al., 2019) $(F_{\text{score}}^{0.1\%}$ and $F_{\text{score}}^{1\%})$. These scores assess the quality of the completion results by comparing them against the ground-truth completions in the test dataset.

**Real Data Completion** The primary advantage of unpaired point cloud completion is its ability to train models using incomplete and completion point cloud data from different sources. In this respect, USSPA (Ma et al., 2023) introduced a benchmark, where incomplete point clouds are obtained from real scans, while complete point clouds come from synthetic data. This dataset consists of ten object categories, including chairs, trash bins, lamps, etc. We evaluate our model in two settings: the **single-category** setting, where performance is assessed separately for each object category, and the **multi-category** setting, where the model is trained and tested across all categories combined. The multi-category setting is particularly challenging, as the model should learn to complete partial point clouds from diverse categories. Our UOT-UPC is compared to diverse paired (*PoinTr, Disp3D, and TopNet*) and unpaired models (*ShapeInv, Unpaired, Cylce4, and USSPA*). For the paired approaches, each model is trained using the paired label provided in the USSPA dataset.

Fig. 2 illustrates completion samples in the single category setting (See Appendix C.1 for qualitative results in the multi-category setting). Table 3 and 4 presents the L1 Chamfer distance $(cd^{l1})$ and F-scores $(F_{\text{score}}^{0.1\%}$ and $F_{\text{score}}^{1\%})$. The AVG column in Table 3 indicates the average $cd^{l1}$ scores across all ten categories. Across almost all settings and metrics, our

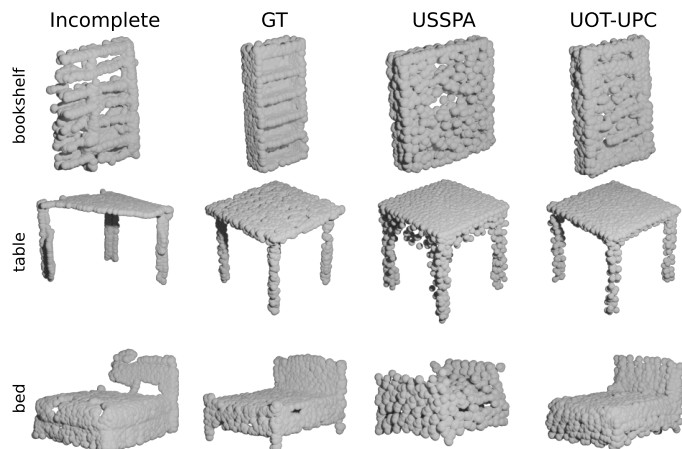

Figure 2: **Comparison of generated samples** from UOT-UPC and USSPA on USSPA dataset in the single-category setting.

Table 3: **Point cloud completion comparison on USSPA dataset in the single-category setting**, assessed by L1 Chamfer Distance ($cd^{l1} \times 10^2$ ($\downarrow$)). The boldface denotes the best performance among unpaired methods. All scores are taken from (Ma et al., 2023).

| | Method | AVG | chair | table | trash bin | TV | cabinet | bookshelf | sofa | lamp | bed | tub |
|---|---|---|---|---|---|---|---|---|---|---|---|---|
| Paired | PoinTr (Yu et al., 2021) | 14.37 | 13.65 | 12.52 | 15.26 | 12.69 | 17.32 | 13.99 | 12.36 | 17.05 | 15.13 | 13.77 |
| | Disp3D (Wang et al., 2022) | 7.78 | 6.24 | 8.20 | 7.12 | 7.12 | 10.36 | 6.94 | 5.60 | 14.03 | 6.90 | 5.32 |
| | TopNet (Tchapmi et al., 2019) | 7.07 | 6.39 | 5.79 | 7.40 | 6.26 | 8.37 | 7.02 | 5.94 | 8.50 | 7.81 | 7.25 |
| Unpaired | ShapeInv (Zhang et al., 2021) | 21.39 | 17.97 | 17.28 | 33.51 | 15.69 | 26.26 | 25.51 | 14.28 | 16.69 | 32.33 | 14.43 |
| | Unpaired (Chen et al., 2020) | 10.47 | 8.41 | 7.52 | 12.08 | 6.72 | 17.45 | 9.95 | 6.92 | 19.36 | 10.04 | 6.22 |
| | Cycle4 (Wen et al., 2021) | 11.53 | 9.11 | 11.35 | 11.93 | 8.40 | 15.47 | 12.51 | 10.63 | 12.25 | 15.73 | 7.92 |
| | USSPA (Ma et al., 2023) | 8.56 | 8.22 | 7.68 | 10.36 | 7.66 | **10.77** | 7.84 | **6.14** | 11.93 | **8.20** | 6.75 |
| | UOT-UPC (Ours) | **7.60** | **7.51** | **6.33** | **8.83** | **6.07** | 11.54 | **7.32** | 6.61 | **7.30** | 9.00 | **5.45** |

Table 4: **Point cloud completion comparison** on USSPA dataset in the single-category setting and the multi-category setting, assessed by L1 Chamfer Distance ($cd^{l1} \times 10^2$ ($\downarrow$)) and F-scores ($F_{\text{score}}^{0.1\%} \times 10^2$, $F_{\text{score}}^{1\%} \times 10^2$ ($\uparrow$)).

| | Method | Single-category (AVG) | | Multi-category | | |
|---|---|---|---|---|---|---|
| | | $F_{\text{score}}^{0.1\%}$ ↑ | $F_{\text{score}}^{1\%}$ ↑ | $cd^{l1}$ ↓ | $F_{\text{score}}^{0.1\%}$ ↑ | $F_{\text{score}}^{1\%}$ ↑ |
| Paired | PoinTr | - | - | 14.37 | 18.35 | 80.41 |
| | Disp3D | - | - | 7.78 | 30.29 | 78.26 |
| | TopNet | - | - | 7.07 | 12.33 | 80.37 |
| Unpaired | ShapeInv | 15.58 | 66.53 | 19.35 | 16.98 | 69.66 |
| | Unpaired | 12.20 | 64.33 | 10.12 | 10.86 | 66.68 |
| | Cycle4 | 9.98 | 60.14 | 12.00 | 8.61 | 56.57 |
| | USSPA | 17.49 | 73.41 | **8.96** | 16.88 | **72.31** |
| | UOT-UPC | **18.43** | **75.59** | **8.96** | **19.25** | 71.52 |

Table 5: **Point cloud completion comparison** on the PCN dataset in the single-category setting, assessed by L1 Chamfer Distance ($cd^{l1} \times 10^2$ ($\downarrow$)).

| | Method | AVG | chair | table | cabinet | sofa | lamp |
|---|---|---|---|---|---|---|---|
| Paired | PoinTr | 5.49 | 5.61 | 5.68 | 6.08 | 5.67 | 4.44 |
| | Disp3D | 2.51 | 2.42 | 2.30 | 2.38 | 2.44 | 3.00 |
| | TopNet | 5.92 | 6.34 | 5.45 | 6.06 | 5.80 | 5.95 |
| Unpaired | ShapeInv | 19.05 | 23.18 | 15.66 | 17.14 | 22.85 | 16.40 |
| | Unpaired | 14.87 | 12.87 | 8.14 | 14.30 | 18.23 | 20.82 |
| | Cycle4 | 17.60 | 14.25 | 15.73 | 21.06 | 21.54 | 15.40 |
| | USSPA | 12.63 | 13.52 | 9.66 | 8.89 | 15.51 | 15.57 |
| | UOT-UPC | **7.87** | **9.96** | **8.74** | **6.41** | **7.83** | **6.42** |

model outperforms other unpaired models. In particular, in the single-category setting, our model attains the best results across all metrics in the AVG column. In the multi-category setting, our model achieves the best performance in $cd^{l1}$ and $F_{\text{score}}^{0.1\%}$, while showing comparable results in $F_{\text{score}}^{1\%}$ (See Appendix C.2 for qualitative results on the real-world KITTI dataset (Geiger et al., 2012)).

**Synthetic Data Completion** We also evaluate our UOT-UPC on the PCN dataset (Yuan et al., 2018). Similar to the USSPA dataset, we compare our model with the same paired and unpaired approaches. Table 5 reports the results for the

single-category setting. Our model significantly outperforms other unpaired approaches, such as USSPA (Ma et al., 2023). Note that the unpaired approaches tackle a more challenging problem, as they do not rely on the pair information between incomplete and complete point clouds. As a result, paired approaches generally achieve higher scores compared to unpaired methods. Nevertheless, our UOT-UPC model achieves competitive performance.

### 5.2. Robustness to Class Imbalance of UOT approach

In this section, we explore the robustness of our model in class-imbalanced settings. Specifically, we examine how the performance of existing point cloud completion models

Table 6: **Comparison of class imbalance robustness** ($cd^{l1} \times 10^2$ (↓)) on (Data1, Data2) = (TV, Table).

| $r$ | 0.3 | 0.5 | 0.7 | 1 |
|---|---|---|---|---|
| USSPA | 7.60 | 6.97 | 8.08 | 7.97 |
| OT-UPC | 23.77 | 23.74 | 29.79 | 27.21 |
| Ours | **6.86** | **6.75** | **6.75** | **6.94** |

Table 7: **Ablation study on the cost** $c(\cdot, \cdot)$ ($cd^{l1} \times 10^2$ (↓)).

| Cost function | Multi-category | trash bin | TV |
|---|---|---|---|
| $l_2$ | 24.16 | 45.57 | 23.71 |
| $cd^{l2}$ | 10.12 | 10.40 | 6.47 |
| $cd^{l2}{}_{fwd}$ | 13.58 | 10.16 | 7.39 |
| InfoCD | **8.96** | **8.83** | **6.07** |

changes with different class imbalance ratios. We select two categories of datasets: Data1 (category: TV) and Data2 (category: Table). These categories are selected because of their relatively abundant training samples and the distinct differences in their shape. For the incomplete point cloud samples, we use the entire training data for both Data 1 and Data2, maintaining their ratio of $6.4 : 21.3$ in Table 2. For the complete point cloud samples, **we manipulate the imbalance ratio** $r$, i.e., Data1 and Data2 are sampled at a ratio of $6.4 : 21.3 \times r$.

$$(\text{Data1} : \text{Data2}) = (6.4 : 21.3) \rightarrow (6.4 : 21.3 \times r).$$

Then, each model is evaluated across diverse values of $r$ to explore the effects of class imbalance. We compare our model to (i) the OT counterpart of our model (OT-UPC) and (ii) USSPA, the state-of-the-art method for unpaired point cloud completion. Note that, as discussed in Sec. 4.2, our model reduces to the OT counterpart when $\Psi_i^* = Id$. For detailed hyperparameter settings, See Appendix A.

As shown in Table 6, our model outperforms the two alternative models across various class imbalance settings. (See Table 9 in the Appendix for results on other class combinations.) Note that we tested $r \leq 1$, because Data2 has a significantly larger total number of training samples, more than three times that of Data1 (Table 2). Hence, setting $r > 1$ would result in discarding too many training data samples. Our model consistently demonstrates stable performance, ranging between 6.75 and 6.94 across various class imbalance ratios $r$, while USSPA shows considerably greater variance. In contrast, the standard OT generally performs poorly. We hypothesize that this phenomenon occurs due to the unstable training dynamics of the standard OT. The stable training dynamics in learning the transport map is also another advantage of the UOT over OT (Choi et al., 2024b). In summary, these results indicate that our UOT-UPC offers strong robustness to class imbalance.

Table 8: **Ablation study on the source mixture trick**, i.e., the complete input.

| Category | Complete Input | $cd^{l1}$ ↓ | $F_{\text{score}}^{0.1\%}$ ↑ | $F_{\text{score}}^{1\%}$ ↑ |
|---|---|---|---|---|
| Single | | 7.94 | 18.12 | 73.49 |
| | ✓ | **7.60** | **18.43** | **75.59** |
| Multi | | 8.98 | 18.04 | **71.75** |
| | ✓ | **8.96** | **19.25** | 71.52 |

### 5.3. Ablation Study

**Effect of Appropriate Cost Functional**   We conduct an ablation study by modifying the cost function $c(\cdot, \cdot)$ in our model (Eq. 8). Each model is evaluated in the multi-category setting and in the single-category settings for the 'trash bin' and 'TV' classes. Table 7 demonstrates that our model achieves the best performance using the InfoCD cost function, followed by ($cd^{l2}_{fwd}$, $cd^{l2}$), and $l^2$ (See Table 10 for the cost ablation results on the PCN dataset). Note that this ranking closely aligns with our cost function analysis in Table 1. This consistency suggests a strong correlation between our theoretical cost function evaluation (Table 1) and actual model performance. Furthermore, these findings suggest that further exploration of alternative cost functions could potentially enhance our model's performance. We leave this exploration for future work.

**Add Complete Sample to Source**   As described in Sec 4.3, we introduce the source mixture trick into our model, where the source distribution is given as a mixture of incomplete and complete point cloud data with a $50\%$ mixing probability. Here, we conduct an ablation study to evaluate the effect of this source mixture trick. The results are presented in Table 8. Across both single-category and multi-category experiments, our model almost consistently exhibits improvements in both $cd^{l1}$ and F scores when utilizing the source mixture trick. Intuitively, this trick aims to assist the transport map in generating the target distribution better. Specifically, when given complete point cloud inputs, the optimal transport map should ideally learn an identity mapping, which is relatively easier compared to completing an input incomplete point cloud. We hypothesize this property encourages the training process, enabling the model to generate higher-fidelity complete point clouds more efficiently.

### 6. Conclusion

In this paper, we introduced UOT-UPC, an unpaired point cloud completion model based on the UOT map. We formulated the unpaired point cloud completion task as the (Unbalanced) OT problem and investigated the cost function for this task. Our experiments demonstrated a strong correlation between cost function selection and completion performance. Our UOT-UPC attains competitive performance compared to both unpaired and paired point cloud completion models. Moreover, our experiments showed that UOT-

UPC presents robustness to the class imbalance problem, which is prevalent in the unpaired point cloud completion tasks. One limitation of our work is that while we explored various candidate cost functions, there may exist better cost functions for this task, e.g., parametrized using neural networks. We leave this exploration for future work. Also, our model sometimes exhibits unstable training dynamics due to adversarial training, which is commonly observed in GAN models.

## Acknowledgements

This work was supported by the NRF grant [RS-2024-00421203] and the IITP grant funded by the Korea government (MSIT) [RS-2021-II211343-GSAI, Artificial Intelligence Graduate School Program (Seoul National University)]. Jaewoong was supported by the National Research Foundation of Korea (NRF) grant funded by the Korea government (MSIT) [RS-2024-00349646]. Jaemoo is supported by National Research Foundation of Korea (NRF) grant funded by the Korea government (MSIT)(RS-2024-00342044).

## Impact Statement

The point cloud completion research contributes positively to various fields, including autonomous driving, robotics and virtual/augmented reality. Also, it is applicable to urban planning and cultural heritage preservation. Our research does not involve personal data or human subjects, and we have carefully addressed potential data bias issues. We also ensure that there are no risks related to illegal surveillance or privacy violations. As a result, we believe that this research is conducted ethically and poses no social or ethical concerns.

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

# A. Implementation Details

Unless otherwise stated, our implementation follows the experimental settings and hyperparameters of USSPA (Ma et al., 2023).

## A.1. Network

We adopt the generator and discriminator architectures from the USSPA framework as completion model $T_\theta$ and potential $v_\phi$. For the potential $v_\phi$, the final sigmoid layer of the discriminator is omitted to allow for the parameterization of the potential function, enabling outputs to assume any real values. Additionally, we remove the feature discriminator to streamline the architecture. In the potential $v_\phi$, we implement the encoder proposed by (Yuan et al., 2018) in their Point Cloud Networks (PCN). Following the encoder, we employ an MLPConv layer specified as $\text{MLPConv}(C_{in}, [C_1, \ldots, C_n]) = \text{MLPConv}(1024, [256, 256, 128, 128, 1])$, which indicates that the output $y$ is computed as follows:

$$y = \text{Conv1D}_{C_4=128, C_5=1}(\text{ReLU}(\ldots \text{ReLU}(\text{Conv1D}_{C_{in}=1024, C_1=256}(x))\ldots)) \tag{9}$$

Here, $\text{Conv1D}_{C_{\text{in}}, C_{\text{out}}}$ represents a 1D convolutional layer with $C_{\text{in}}$ input channels and $C_{\text{out}}$ output channels.

The completion model $T_\theta$ receives as input a concatenation of the incomplete point cloud and a complete point cloud. These inputs are processed independently to generate distinct complete point cloud samples. The completion model $T_\theta$ follows an Encoder-Decoder architecture, augmented by an upsampling refinement module (upsampling module) in sequence. The upsampling module is implemented using a 4-layer MLPConv network, where the final MLPConv layer is responsible for refining and adding detailed structures to the output (Ma et al., 2023). Specifically, the inputs to the last MLPConv layer are composed of the skeleton point cloud produced by the Encoder-Decoder structure and the features extracted from the third MLPConv layer.

## A.2. Implementation detail

**Motivation - Optimal Cost Function**  The incomplete and complete point clouds utilized in the optimal cost function outlined in Sec 4.1 are sourced from the dataset proposed by (Ma et al., 2023). This dataset consists of paired incomplete and complete point clouds. For a fair comparison, we shuffle the complete point clouds to create an unpaired setting. We then use these shuffled point clouds as artificial complete data to train the USSPA model.

**Training**  Concerning the loss function $L_{v,T}$. We employ Infocd as the cost function $c$ with a coordinate value of $\tau = 0.044$. For the hyperparameters of InfoCD, we set $\tau_{\text{infocd}}$ to 2 and $\lambda_{\text{InfoCD}}$ to $1.0 \times 10^{-7}$. The functions $\Psi_1^*$ and $\Psi_2^*$ are defined using the Softplus activation, $\text{SP}(x) = 2\log(1 + e^x) - 2\log 2$.[2] As a regularization term, we incorporate the density loss $dl$ proposed by (Ma et al., 2023), and we designate a coordinate value of 10.5 for $dl$. The objective of Potential $v_\phi$ is to assign high value to target sample $y$ while assigning lower values to generated sample $\hat{y}$. We utilize the Adam optimizer with $\beta_1 = 0.95, \beta_2 = 0.999$ and learning rates of $1.0 \times 10^{-5}$ for both the potential $v_\phi$ and completion model $T_\theta$, respectively. The training is conducted with a batch size 4. The maximum epoch of training is 480. We report the final results based on the epoch that yields the best performance.

**Ablation study - Effect of Appropriate Cost Functional**  We set cost function coordinate value $\tau = 100$ for cost function $cd^{l2}{}_{fwd}, cd^{l2}$ and $l^2$.

**Ablation study - Add Complete Sample to Source**  For a fair comparison, we set the learning rates of $2.0 \times 10^{-5}$ for both the potential $v_\phi$ and completion model $T_\theta$, respectively, when not using the source mixture trick.

---

[2] The softplus function is translated and scaled to satisfy $\text{SP}(0) = 0$ and $\text{SP}'(0) = 1$.

**Evaluation Metrics**

- $L1$-Chamfer Distance $cd^{l1}$ (Fan et al., 2017)

$$cd^{l1}(x_i, y_j) = \frac{1}{2}\left(\frac{1}{|x_i|}\sum_m \min_n \|x_{im} - y_{jn}\|_2 + \frac{1}{|y_j|}\sum_n \min_m \|x_{im} - y_{jn}\|_2.\right) \tag{10}$$

where each of $x_i, y_j$ is point cloud

- $F$ score $F^\alpha_{score}$ (Tatarchenko et al., 2019)

$$F^\alpha_{score} = \frac{2 \times P(\alpha) \times R(\alpha)}{P(\alpha) + R(\alpha)} \tag{11}$$

where $P(\alpha) = \frac{|\{x_{im} \in x_i\,|\,\min_n(\|x_{im} - y_{jn}\|_2) < \alpha\}|}{|x_i|}$ measures the accuracy of $x_i$,

and $R(\alpha) = \frac{|\{y_{jn} \in y_j\,|\,\min_m(\|x_{im} - y_{jn}\|_2) < \alpha\}|}{|y_j|}$ measures the completeness of $x_i$.

### A.3. OT-UPC

For the potential $v_\phi$, we implement $\text{MLPConv}(512, [128, 128, 1])$ following the PCN encoder (Yuan et al., 2018). We incorporate R1 regularization (Roth et al., 2017) and R2 regularization (Mescheder et al., 2018) to the loss function $L_{v,T}$. Both regularization terms are assigned coordinate values $r1 = r2 = 0.2$. The density loss $dl$ is excluded from the $L_{v,T}$. A gradient clipping value of 1.0 is applied. We use Adam optimizer with $\beta_1 = 0.9, \beta_2 = 0.999$ and a learning rate $lr_{T_\theta} = 5.0 \times 10^{-5}$ for the completion model $T_\theta$. In addition, we use Adam optimizer with $\beta_1 = 0.9, \beta_2 = 0.999$ and learning rate $lr_{v_\phi} = 1.0 \times 10^{-7}$ for the potential $v_\phi$. All other settings not explicitly mentioned follow those of our model, UOT-UPC.

# B. Cost Function Evaluation

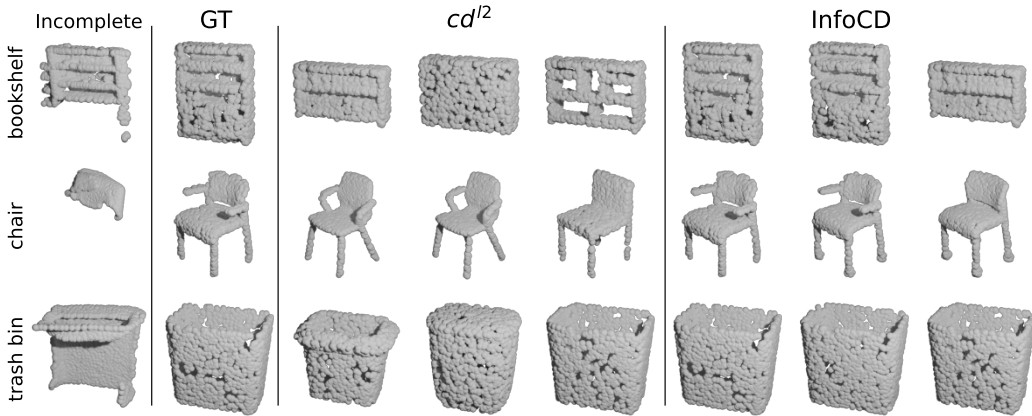

Figure 3: **Visualization of the incomplete point cloud** $x$**, the ground-truth completion** $y^{gt}(x)$**, and the three complete point clouds** $y_i^c(x)$ that minimize the cost $c(x, y_i^c(x)$ for two cost functions: $cd^{l2}$ and InfoCD, in the **single-category setting**.

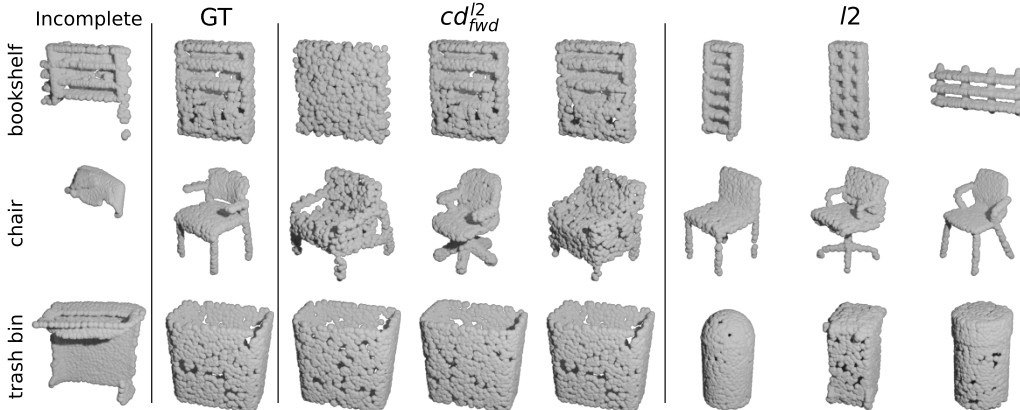

Figure 4: **Visualization of the incomplete point cloud** $x$**, the ground-truth completion** $y^{gt}(x)$**, and the three complete point clouds** $y_i^c(x)$ that minimize the cost $c(x, y_i^c(x))$ for two cost functions: $cd^{l2}{}_{fwd}$ and $l2$, in the **single-category setting**.

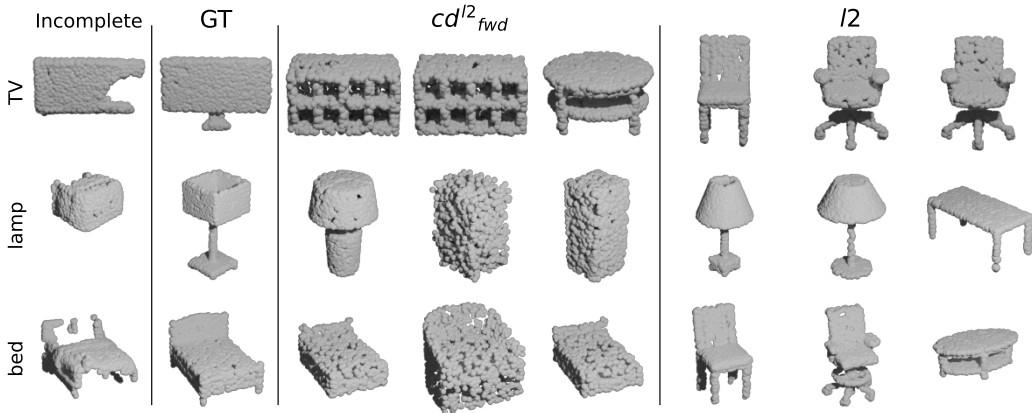

Figure 5: **Visualization of the incomplete point cloud** $x$**, the ground-truth completion** $y^{gt}(x)$**, and the three complete point clouds** $y_i^c(x)$ that minimize the cost $c(x, y_i^c(x))$ for two cost functions: $cd^{l2}_{fwd}$ and $l2$, in the **multi-category setting**.

# C. Additional Results

## C.1. Additional Qualitative Results

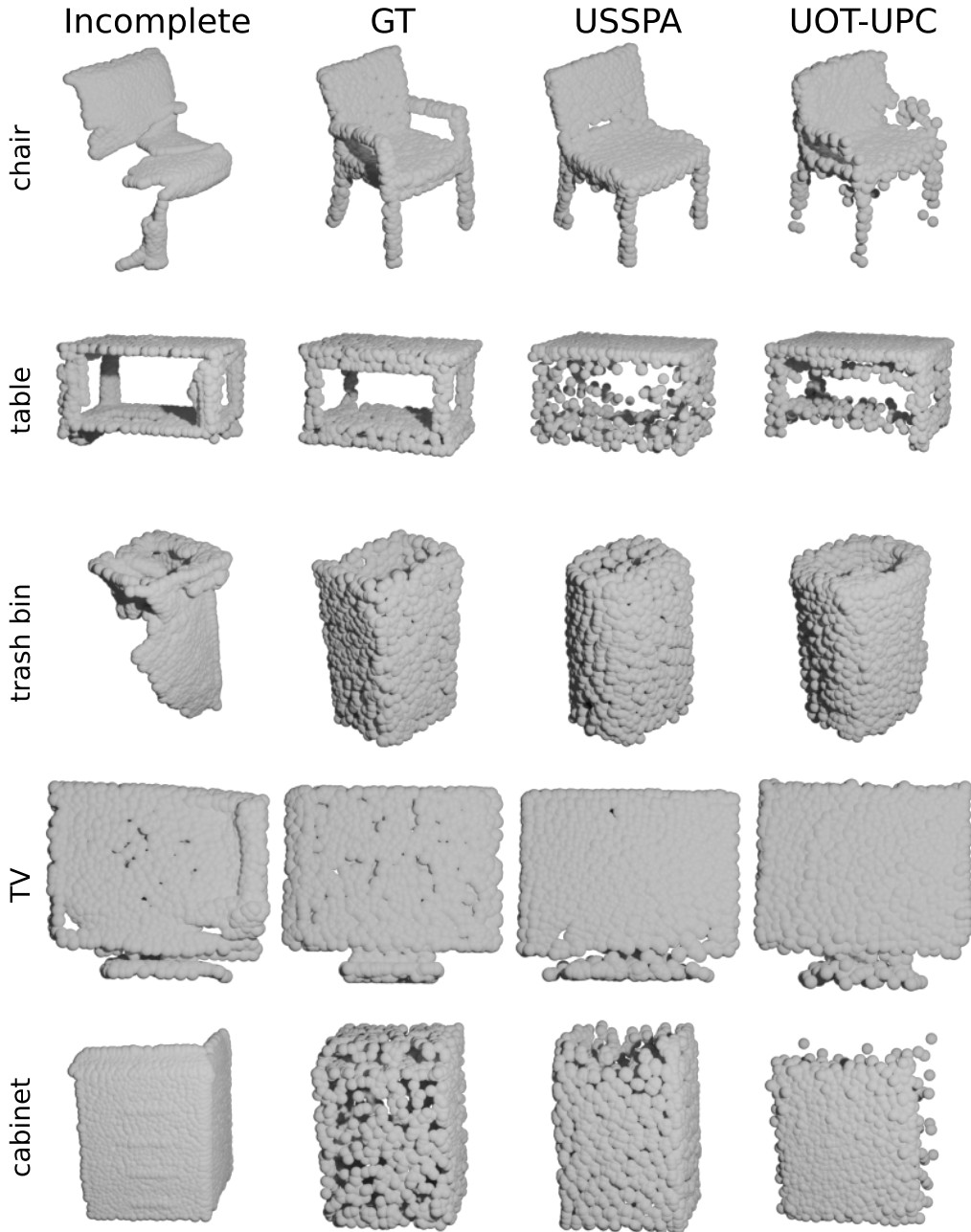

Figure 6: **Comparison of generated samples** from our UOT-UPC and USSPA in the single-category setting.

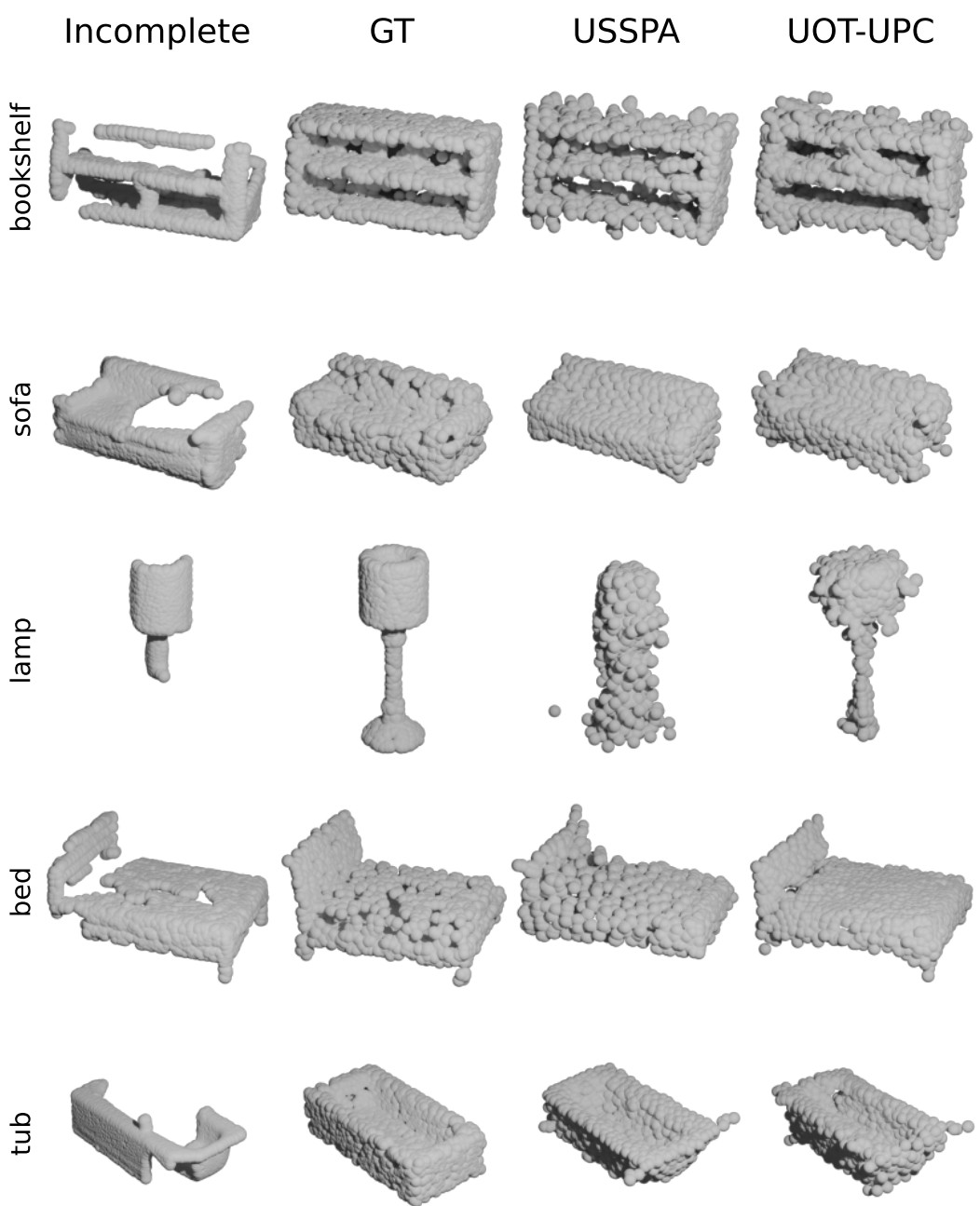

Figure 7: **Comparison of generated samples** from our UOT-UPC and USSPA in the single-category setting.

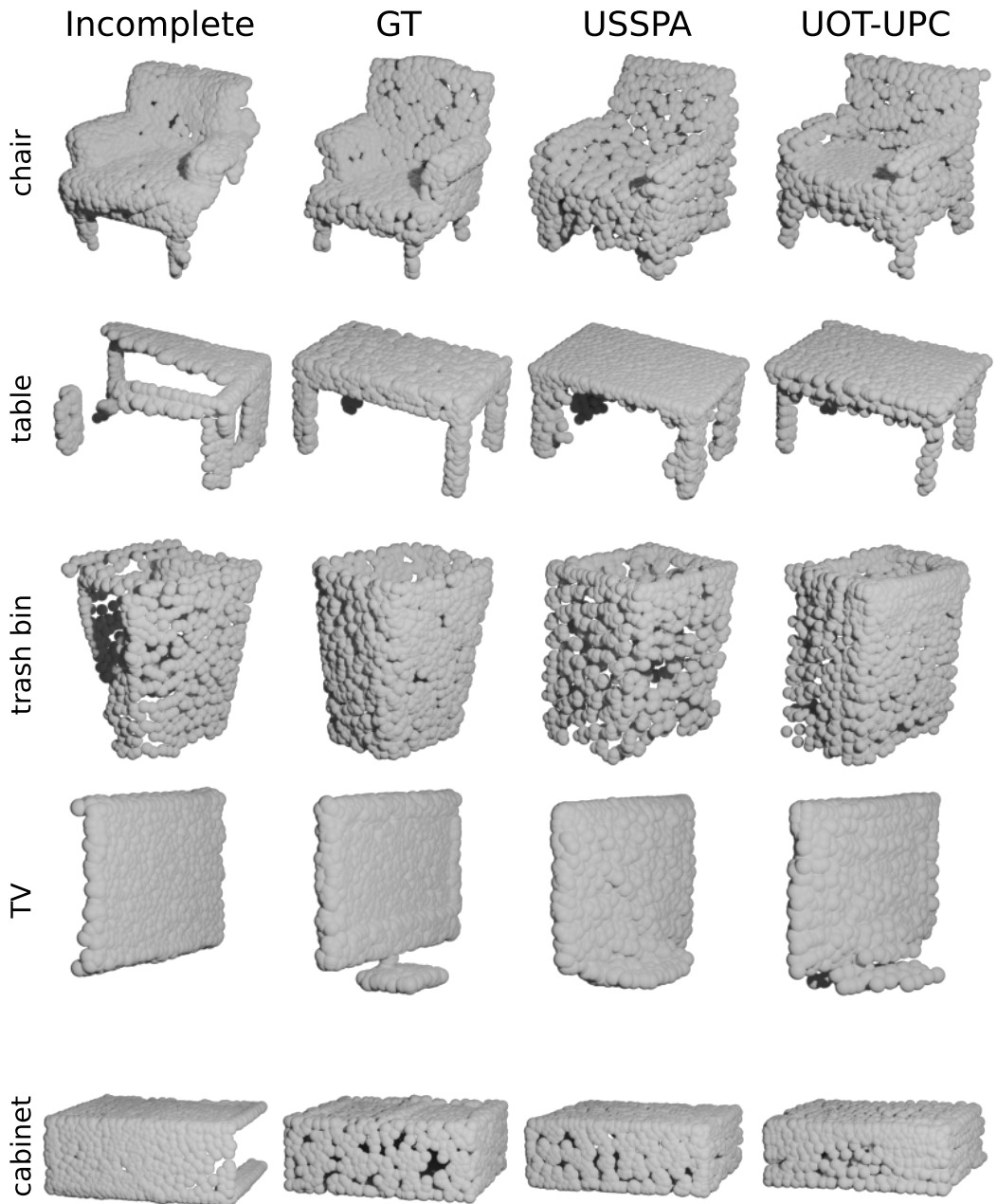

Figure 8: **Comparison of generated samples** from our UOT-UPC and USSPA in the multi-category setting.

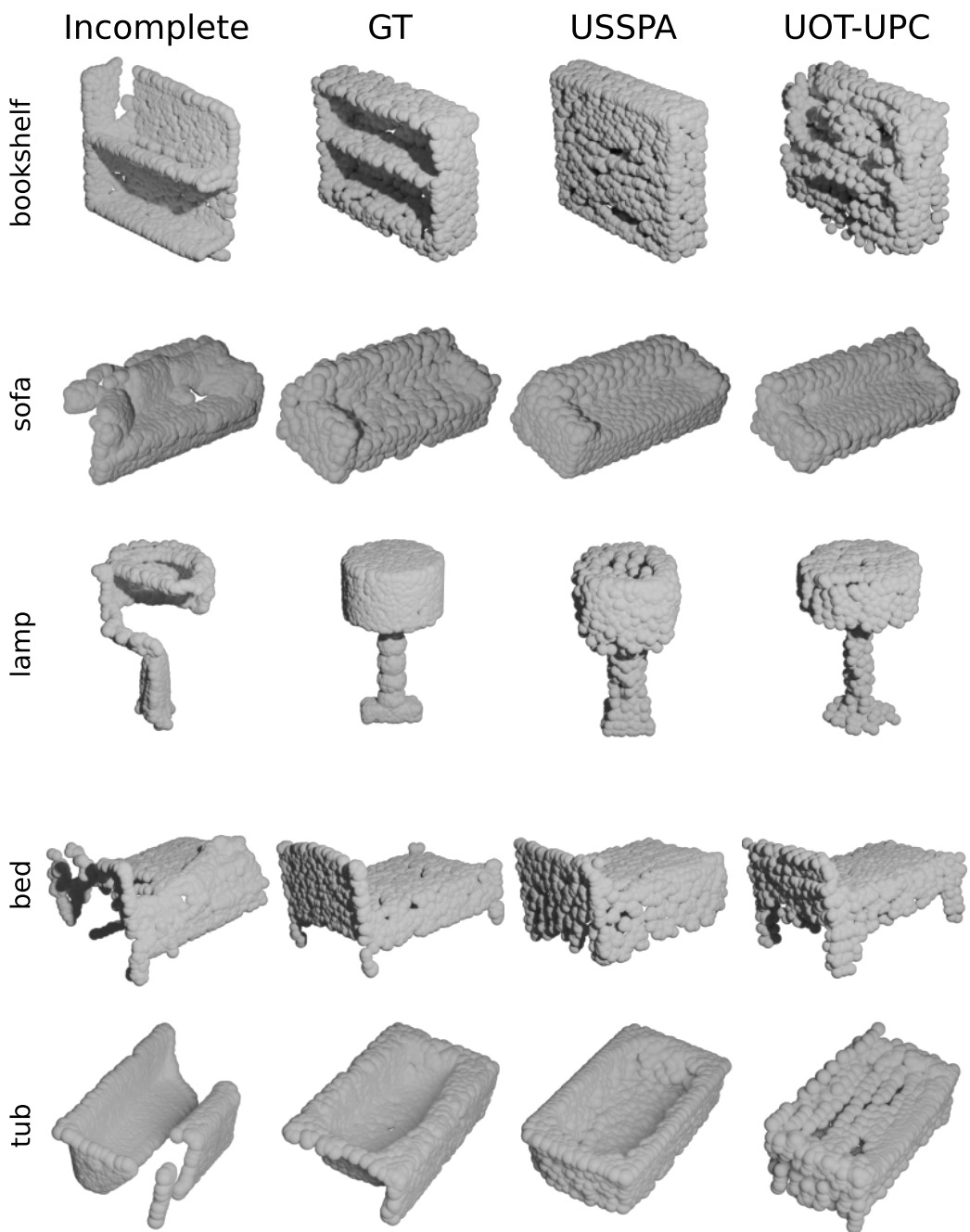

Figure 9: **Comparison of generated samples** from our UOT-UPC and USSPA in the multi-category setting.

**C.2. Qualitative comparison between our UOT-UPC and existing methods on the KITTI dataset.**

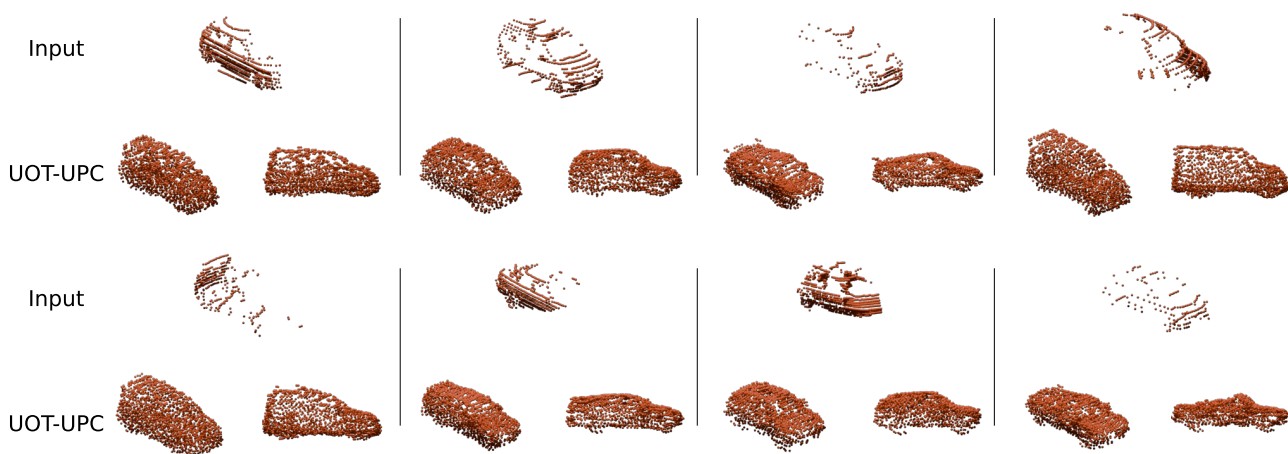

Figure 10: **Point cloud completion results of the UOT-UPC model on the KITTI dataset (Geiger et al., 2012)**. The model is trained on the ShapeNet dataset under the car category and tested on partial point clouds from the KITTI dataset without fine-tuning. From the qualitative comparison with previous approaches (Fig 11), our UOT-UPC model achieves higher-fidelity point cloud completion, demonstrating better global structure and more evenly distributed points.

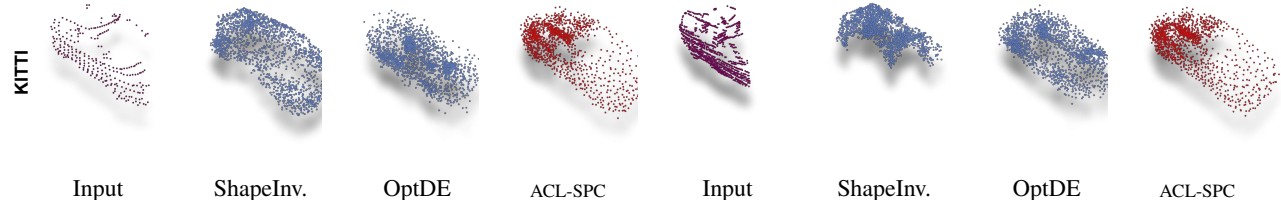

Figure 11: **Point cloud completion results of previous models on the KITTI dataset (Geiger et al., 2012)**. The generated samples are taken from ACL-SPC (Hong et al., 2023), which is a self-supervised model. The others are unsupervised approaches: ShapeInv (Zhang et al., 2021) and OptDE (Gong et al., 2022).

## C.3. Comparison of class imbalance robustness for diverse class combinations.

Table 9: **Comparison of class imbalance robustness** ($cd^{l1} \times 10^2$ ($\downarrow$)) between UOT-UPC (ours), USSPA, and OT-UPC on diverse class combinations (Data1, Data2). Our UOT-UPC consistently outperforms other models across a wide range of class imbalance ratios in both additional class settings.

(a) (Data1, Data2) = (Lamp, Trash bin) with sample count = (1.1 : 8.0 * $r$).

| $r$ | 0.3 | 0.5 | 0.7 | 1 |
|---|---|---|---|---|
| USSPA | 10.16 | 9.49 | 10.21 | 10.21 |
| OT | 25.68 | 21.95 | 28.41 | 25.36 |
| Ours | **9.42** | **9.48** | **9.57** | **9.44** |

(b) (Data1, Data2) = (Lamp, Bed) with sample count = (1.1 : 2.9 * $r$).

| $r$ | 0.3 | 0.5 | 0.7 | 1 |
|---|---|---|---|---|
| USSPA | 9.64 | 9.78 | 9.27 | 9.79 |
| OT | 18.99 | 21.23 | 19.27 | 22.12 |
| Ours | **8.95** | **8.91** | **8.98** | **8.73** |

## C.4. Additional experimental results on the PCN dataset

Table 10: **Ablation study on the cost function** $c(\cdot, \cdot)$ on the PCN dataset ($cd^{l1} \times 10^2$ ($\downarrow$)). The results are consistent with Table 7. InfoCD achieved the best performance, while the L2 distance yielded the worst results.

| Cost function | cabinet | sofa | lamp |
|---|---|---|---|
| $l_2$ | 23.86 | 19.92 | 19.22 |
| $cd^{l2}$ | 8.85 | 8.40 | 7.07 |
| $cd^{l2}{}_{fwd}$ | 13.53 | 11.32 | 12.78 |
| InfoCD | **6.41** | **7.83** | **6.42** |

### C.5. Ablation study on cost-intensity $\tau$

We evaluate the robustness of our model with respect to the cost-intensity hyperparameter $\tau$, defined as $c(x,y) = \tau \times$ InfoCD$(x,y)$. Specifically, we tested our model on the multi-category setting and the single-category settings of the 'bookshelf' and 'lamp' classes, while changing $\tau \in \{0.02, 0.025, 0.044, 0.1, 0.25\}$. Note that we impose challenging conditions by setting the maximum $\tau$ to $\tau_{\max} = 0.25$ and the minimum $\tau$ to $\tau_{\min} = 0.02$, resulting in a ratio of $\tau_{\max}/\tau_{\min} > 10$. As depicted in Fig. 12, our model shows moderate performance across various $\tau$ values. In particular, the sweet spot of $\tau$ lies roughly between 0.044 and 0.1. The performance deteriorates by approximately 10% when $\tau$ is either too large ($\tau_{\max}$) or too small ($\tau_{\min}$).

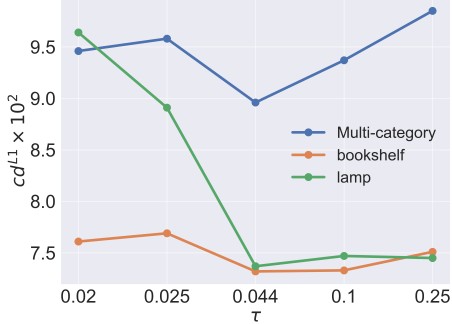

Figure 12: **Ablation study on the cost intensity** $\tau$ ($cd^{l1} \times 10^2$ ($\downarrow$)).

