# OpenReview forum: "Unpaired Point Cloud Completion via Unbalanced Optimal Transport"
_ICML.cc/2025/Conference — ICML 2025 poster_

### Official Review · Reviewer_LXHR · 2025-03-13

**Overall Recommendation:** 4

**Summary:**

This paper proposes UOT-UPC, a novel approach to unpaired point cloud completion using the Unbalanced Optimal Transport framework. The model formulates the completion task as an optimal transport problem and trains a neural network-based Neural OT map to learn the transport mapping from incomplete to complete point clouds. UOT-UPC is designed to handle class imbalance, a common issue in real-world unpaired completion tasks.

**Claims And Evidence:**

Yes

**Essential References Not Discussed:**

No

**Experimental Designs Or Analyses:**

The proposed method was evaluated on the USSPA and PCN datasets, where it demonstrated superior performance compared to previous state-of-the-art methods. Extensive ablation studies were conducted to thoroughly analyze the impact of different cost functions, further validating the effectiveness of the proposed approach.

**Methods And Evaluation Criteria:**

The proposed methods are well-aligned with the problem of unpaired point cloud completion. The paper formulates unpaired point cloud completion as an Optimal Transport (OT) problem, which is theoretically sound for aligning distributions between incomplete and complete point clouds. The introduction of Unbalanced Optimal Transport (UOT) is particularly meaningful, as real-world point cloud datasets often exhibit class imbalance, making traditional OT less effective.

**Other Comments Or Suggestions:**

The results of the proposed approach should be added to Figure 11.

**Other Strengths And Weaknesses:**

**Strengths:**

- First work using UOT for unpaired point cloud completion.
- Strong experimental validation across multiple datasets.
- Addresses class imbalance problem, a major issue in real-world data.

**Weaknesses:**

- Training instability due to adversarial training aspects of OT.

**Questions For Authors:**

Why is InfoCD the most suitable for completion tasks? Please provide a theoretical explanation. MSNet and SpareNet previously used EMD to calculate the distance between point sets. Would EMD based on optimal transport be better than InfoCD?

**Relation To Broader Scientific Literature:**

Prior unpaired completion methods (Cycle4 (Wen et al., 2021), USSPA (Ma et al., 2023)) use heuristic-driven adversarial training. The paper reformulates unpaired point cloud completion as an UOT problem which bridges OT theory and unpaired point cloud completion, addressing limitations in existing heuristic-driven methods.

**Theoretical Claims:**

The paper presents theoretical claims related to the OT and UOT formulation for unpaired point cloud completion. The proofs supporting these theoretical claims are mathematically sound. The paper conducts a comprehensive analysis of various cost functions and concludes that InfoCD is the most suitable for completion tasks. This conclusion is primarily supported by experimental validation rather than additional mathematical proofs. The paper acknowledges that the UOT-UPC training process may exhibit instability, resembling the mode collapse phenomenon observed in GAN training. However, no theoretical analysis is provided to explain why UOT-UPC training might be unstable.

---

> ### Author Rebuttal · Authors · 2025-03-29
>
> We sincerely thank the reviewer for carefully reading our manuscript and providing valuable feedback. Moreover, we appreciate the reviewer for considering our work addresses "limitations in existing heuristic-driven methods" by "bridging OT theory and unpaired point cloud completion". We hope our responses to be helpful in addressing the reviewer's concerns.
>
> $ $
>
> ---
> > **Q.[Questions For Authors]** Why is InfoCD the most suitable for completion tasks? Please provide a theoretical explanation. MSNet and SpareNet previously used EMD to calculate the distance between point sets. Would EMD based on optimal transport be better than InfoCD?
>
> **A.** Thank you for providing the insightful comments. Following the reviewer's suggestion, **we conducted an additional ablation study using the EMD as the cost function for our UOT-UPC model**. The results are as follows (Table 7):
>
> - Comparison of EMD cost function with other cost functions on the USSPA benchmark, assessed by L1 Chamfer Distance $cd^{l 1} \times 10^2$ ($\downarrow$).
>
> |Cost function| Multi-category | trash bin | TV |
> |:---|:---|:---|:---|
> |$l_{2}$| 24.16 | 45.57 | 23.71|
> |$cd^{l2}$| 10.12 | 10.40 | 6.47|
> |$cd^{l2}_{fwd}$| 13.58 | 10.16 | 7.39|
> |EMD| 9.66 | 10.46 | 6.41|
> |Ours(InfoCD)| **8.96** | **8.83** | **6.07** |
>
> Our results show that **InfoCD consistently outperforms EMD** across all evaluated categories. Additionally, **training with EMD incurs significantly higher computational costs** due to its iterative computation procedure, as shown below:
>
> - Train time comparison on 'lamp' category from the USSPA [1].
>
> |Train (480 Epoch) | Time (sec) |
> |:---|:---|
> |EMD | 3781.91|
> |Ours(InfoCD) | 1320.80 |
>
> InfoCD is designed to prevent multiple points from being matched to a single point, **encouraging a more evenly distributed alignment** between point sets, compared to EMD (Fig 4 in [1]). This advantage arises from the **contrastive nature of the InfoCD** cost function [1], which introduces an additional repulsion effect between negative pairs.
> This property aligns well with the goals of point cloud completion, where generating globally coherent and evenly spaced completions is critical. Empirically, this is supported by the higher-fidelity completions of the UOT-UPC model using InfoCD in Fig  2 and 10, where the **completions exhibit more globally evenly separated completions**.
>
> $ $
>
> Reference
> - [1] Lin, Fangzhou, et al. "InfoCD: a contrastive chamfer distance loss for point cloud completion." NeurIPS 2023.
>
> $ $
>
> ---
> > **Q. [Theoretical Claims]** ... The paper acknowledges that the UOT-UPC training process may exhibit instability, resembling the mode collapse phenomenon observed in GAN training. However, no theoretical analysis is provided to explain why UOT-UPC training might be unstable.
>
>
> **A.** We appreciate the reviewer for providing constructive comments. As the reviewer said, we believe the observed training instability is due to the inherent difficulty of finding a Nash equilibrium in the min-max optimization, similar to challenges encountered in GAN training. In this regard, **several works investigated GAN instability both theoretically [1] and empirically [2]**. We agree that extending this analysis to the UOT-UPC setting would be a valuable future research direction.
>
> Reference
> - [1] Mescheder, Lars, Andreas Geiger, and Sebastian Nowozin. "Which training methods for GANs do actually converge?." ICML 2018
> - [2] Salimans, Tim, et al. "Improved techniques for training gans." NeurIPS 2016.
>
> $ $
>
> ---
> > **Q.[Other Comments Or Suggestions]** The results of the proposed approach should be added to Figure 11.
>
> **A.** We agree with the reviewer that including UOT-UPC results in Fig 11, i.e., providing a completion result on the exact same incomplete point cloud, would allow a clearer qualitative comparison. However, we were unable to find the random seed necessary for reproducibility in the ACL-SPC model. As an alternative, **we chose to present multiple completion results on the KITTI dataset in Fig 10**. Across multiple incomplete point clouds, our model consistently produces higher-fidelity completions, exhibiting better global structure and more evenly distributed points.

---

> > ### Comment · Reviewer_LXHR · 2025-04-06
> >
> > I appreciate the thorough reply from the authors. The majority of my questions have been clarified. In particular, the comparison between the InfoCD and EMD losses was especially helpful.

---

### Official Review · Reviewer_hu8B · 2025-03-13

**Overall Recommendation:** 4

**Summary:**

This paper studies the problem of reconstructing 3d objects from partial observations, an important problem in real-world graphics applications. While some approaches to this problem consider settings where a large dataset of partial and full observations of the same objects are available, this work focuses on the likely more practical settings where one merely has a dataset of partial observations and another dataset of full observations but the observations don't correspond to the same objects. Their method is based on solving an optimal transport problem with neural networks, with the important modification of allowing for un-balancedness, which improves the robustness of their method to imbalances in the class distribution of the data.

**Claims And Evidence:**

Their main claims are that their method is roughly state-of-the-art for unpaired data, and at least competitive for paired data, and also that it is robust to imbalances in the class distribution of the data. These claims are well-supported in my view.

I also appreciated their careful exploration of the relative merits of cost functions as well as the performance of other methods under major class imbalances. Overall, I feel that the paper is very careful and empirical solid.

**Essential References Not Discussed:**

I am not aware of any un-discusses essential references.

**Experimental Designs Or Analyses:**

The experimental designs seem legitimate but I have not made a careful study of them.

**Methods And Evaluation Criteria:**

Yes.

**Other Comments Or Suggestions:**

- You define the "unbalanced optimal transport map" on line 129, but does this necessarily exist?

**Other Strengths And Weaknesses:**

None additional.

**Questions For Authors:**

None.

**Relation To Broader Scientific Literature:**

This paper pushes forward the state of the art of point cloud completion by considering the un-paired problem and designing a strong method based on optimal transport which is also robust to class imbalances.

**Theoretical Claims:**

It is a practical paper so there aren't significant theoretical claims.

---

> ### Author Rebuttal · Authors · 2025-03-29
>
> We sincerely thank the reviewer for carefully reading our manuscript and providing valuable feedback. We are especially grateful for the reviewer’s recognition of our contribution, noting that “this paper pushes forward the state of the art of point cloud completion by considering the unpaired problem and designing a strong method based on optimal transport, which is also robust to class imbalances.” Moreover, we appreciate the reviewer's positive assessment of our cost function analysis and empirical evaluations under class imbalance.
>
> $ $
>
> ---
> > **Q. [Other Comments Or Suggestions]** You define the "unbalanced optimal transport map" on line 129, but does this necessarily exist?
>
> **A.** Thank you for the great question. In this work, we assume that the incomplete and complete point cloud distributions are absolutely continuous with respect to the Lebesgue measure (Lines 76-77). This is a mild assumption, as it is satisfied whenever the distributions admit probability density functions (pdfs). Under this assumption, **the existence of the unbalanced optimal transport map is guaranteed**, as stated in Thm 3.3 in [1].
>
> Reference
> - [1] Liero, Matthias, Alexander Mielke, and Giuseppe Savaré. "Optimal entropy-transport problems and a new Hellinger–Kantorovich distance between positive measures." Inventiones mathematicae 211.3 (2018): 969-1117.

---

> > ### Comment · Reviewer_hu8B · 2025-04-07
> >
> > Ok, could you add this reference to the main text? Thanks!

---

> > > ### Author Response · Authors · 2025-04-07
> > >
> > > Thank you for reviewing our work! We will revise **Line 129** to clarify the existence of the unbalanced optimal transport map and incorporate the suggested reference as follows:
> > > > We refer to the optimal transport map $T^{\star}$ from $\pi_{0}$ to $\pi_{1}$ as the unbalanced optimal transport map (UOT Map). Note that, under our assumption that the source and target distributions are absolutely continuous, the existence of this UOT Map is guaranteed ([1], Thm 3.3).

---

### Official Review · Reviewer_xCQJ · 2025-03-14

**Overall Recommendation:** 3

**Summary:**

This paper introduces Unbalanced Optimal Transport Map for Unpaired Point Cloud Completion (UOT-UPC), which is a novel point cloud completion approach that uses unpaired point clouds during training. Unlike previous approaches that have formulated the point cloud completion task as an optimal transport problem, the authors propose to instead formulate it as an unbalanced optimal transport problem to loosen the exact matching constraint in OT, which helps with training on unbalanced datasets. They train a Neural OT to learn a UOT map that transports incomplete point cloud to complete point cloud using InfoCD as the cost function. The authors evaluate their approach on different, real, synthetic, and hybrid datasets and show impressive performance compared to other models.

**Claims And Evidence:**

The claims are supported by clear and convincing evidence.

**Essential References Not Discussed:**

There are not necessarily related works that are essential to understanding the key contributions which are missing.

**Experimental Designs Or Analyses:**

The experimental designs are sound. The datasets used consist of real and synthetic scans of objects, as well as real world scans which cover a variety of different types of data.

**Methods And Evaluation Criteria:**

The methods and evaluation criteria make sense of this problem.

**Other Comments Or Suggestions:**

The name of the approach is misspelled in the bolded section of the third paragraph in the introduction. Also, the beginning of Section 3.1 defined the set $X$ as $X = \{x_i | x_i \in X, i = 1, \cdots , N\}$ for incomplete point clouds and $Y = \{y_j | y_j \in Y, j = 1, \cdots , M\}$ for complete point clouds. However, the Cost Function Comparison uses $n$ for complete and $m$ for incomplete. I believe it would be a bit clearer to change the variables used for the Task formulation section.

**Other Strengths And Weaknesses:**

This paper introduces an original and well-motivated approach to unpaired point cloud completion by using the unbalanced optimal transport formulation. The authors provide a comprehensive evaluation of their approach through their experiments and ablations, which show impressive performance and robustness on unbalanced datasets. The paper itself is also well written overall, with an algorithm figure that effectively describes their approach. However, the paper's clarity could be improved by explicitly defining important benchmarks before using them in the text. For example, Section 3.1 first compares UOT-UPC with USSPA without first introducing USSPA.

**Questions For Authors:**

No questions so far

**Relation To Broader Scientific Literature:**

There has been a growing research focus on training models with unlabeled or minimally labeled data, particularly in 2D feature representation and unsupervised learning. This paper relates to this as it uses unpaired point cloud data to train a model for point cloud completion, which is unsupervised in nature. Another problem that is investigated is unbalanced datasets, which the paper shows UOT-UPC performs well with. I think that the most important contribution is that this paper is the first to bring UOT to unsupervised point cloud completion, where the problem has been typically formulated as a regular OT problem.

**Theoretical Claims:**

I have not checked the correctness of any proofs for theoretical claims.

---

> ### Author Rebuttal · Authors · 2025-03-29
>
> We sincerely thank the reviewer for carefully reading our manuscript and providing valuable feedback. Moreover, we appreciate the reviewer for considering "the most important contribution is that this paper is the first to bring UOT to unsupervised point cloud completion". We hope our responses to be helpful in addressing the reviewer's concerns.
>
> $ $
>
> ---
> > **Q. [Other Strengths And Weaknesses]** The paper's clarity could be improved by explicitly defining important benchmarks before using them in the text. For example, Section 3.1 first compares UOT-UPC with USSPA without first introducing USSPA.'
>
>
> **A.** We appreciate the reviewer for providing valuable suggestions. Following the reviewer's advice, **we moved the Related Works section (Sec 4) before the Method (Sec 3)**. This reordering allows us to introduce USSPA before presenting experimental results in Section 3.1.
>
> $ $
>
> ---
> > **Q. [Other Comments Or Suggestions]** The name of the approach is misspelled in the bolded section of the third paragraph in the introduction. Also, the beginning of Section 3.1 defined the set $X$ as: $$X = \{x_i \mid x_i \in X, i = 1, \dots, N\}$$ for incomplete point clouds and the set $Y$ as: $$Y = \{y_j \mid y_j \in Y, j = 1, \dots, M\}$$  for complete point clouds. However, the cost function comparison uses $n$ for complete and $m$ for incomplete point clouds. I believe it would be a bit clearer to change the variables used for the Task formulation section.
>
> **A.** We appreciate the reviewer for the careful comment. (1) We corrected the typo in the bolded section of the third paragraph in the introduction. (2) Following the reviewer's advice, we revised the variable notation in the cost function definitions to ensure consistency. Specifically, we now consistently use the $n$ for the incomplete point clouds $x$ and $m$ for the complete point clouds $y$ throughout the paper.

---

### Decision · Program_Chairs · 2025-05-01

**Decision:**

Accept (poster)

**Comment:**

This paper presents a clear and original contribution by introducing Unbalanced Optimal Transport (UOT) to the task of unpaired point cloud completion, a problem of practical significance in 3D vision where paired data is rarely available. The proposed model, UOT-UPC, reformulates completion as a UOT problem and demonstrates strong robustness to class imbalance—an important and realistic challenge in this domain. Reviewers highlighted several strengths: the novelty of applying UOT in this context, the strong empirical performance on both synthetic and real datasets, and the comprehensive ablation studies—particularly on the impact of the cost function choice. The authors also clearly motivated their use of the InfoCD cost and supported it with both quantitative results and theoretical intuition. The paper is well-structured, and the authors were responsive during the rebuttal phase, addressing all technical and presentation-related suggestions (including clarity improvements and additional experimental comparisons). Overall, this work advances the state of the art in unpaired point cloud completion with a well-motivated, well-executed approach grounded in optimal transport theory. I recommend acceptance.